# Detection of Hypertension-Induced Changes in Erythrocytes by SERS Nanosensors

**DOI:** 10.3390/bios12010032

**Published:** 2022-01-08

**Authors:** Evelina I. Nikelshparg, Adil A. Baizhumanov, Zhanna V. Bochkova, Sergey M. Novikov, Dmitry I. Yakubovsky, Aleksey V. Arsenin, Valentyn S. Volkov, Eugene A. Goodilin, Anna A. Semenova, Olga Sosnovtseva, Georgy V. Maksimov, Nadezda A. Brazhe

**Affiliations:** 1Department of Biophysics, Biological Faculty, Moscow State University, 119991 Moscow, Russia; adilbayzhumanov@biophys.msu.ru (A.A.B.); zh.bo4kova@yandex.ru (Z.V.B.); gmaksimov@mail.ru (G.V.M.); 2Center for Photonics and 2D Materials, Moscow Institute of Physics and Technology (MIPT), 141700 Dolgoprudny, Russia; novikov.s@mipt.ru (S.M.N.); dmitrii.yakubovskii@phystech.edu (D.I.Y.); arsenin.av@mipt.ru (A.V.A.); vsv.mipt@gmail.com (V.S.V.); 3Faculty of Materials Sciences, Moscow State University, 119991 Moscow, Russia; goodilin@yandex.ru (E.A.G.); semenovaAA@my.msu.ru (A.A.S.); 4Department of Chemistry, Moscow State University, 119991 Moscow, Russia; 5Department of Biomedical Sciences, Faculty of Health and Medical Sciences, University of Copenhagen, 2200 Copenhagen, Denmark; olga@sund.ku.dk; 6Department of Physical Material Engineering, Federal State Autonomous Educational Institution of Higher Education “National Research Technological University “MISiS”, 119049 Moscow, Russia

**Keywords:** surface-enhanced Raman spectroscopy, hemoglobin, erythrocytes, plasma membrane, biosensing, nanoparticles, plasmonic nanostructures, spontaneously hypertensive rats

## Abstract

Surface-enhanced Raman spectroscopy (SERS) is a promising tool that can be used in the detection of molecular changes triggered by disease development. Cardiovascular diseases (CVDs) are caused by multiple pathologies originating at the cellular level. The identification of these deteriorations can provide a better understanding of CVD mechanisms, and the monitoring of the identified molecular changes can be employed in the development of novel biosensor tools for early diagnostics. We applied plasmonic SERS nanosensors to assess changes in the properties of erythrocytes under normotensive and hypertensive conditions in the animal model. We found that spontaneous hypertension in rats leads (i) to a decrease in the erythrocyte plasma membrane fluidity and (ii) to a decrease in the mobility of the heme of the membrane-bound hemoglobin. We identified SERS parameters that can be used to detect pathological changes in the plasma membrane and submembrane region of erythrocytes.

## 1. Introduction

The development of novel methods and techniques has expanded our understanding of the molecular mechanisms underlying many pathologies and paved the way for the creation of novel diagnostic tools [1,2,3]. Cardiovascular diseases remain the leading cause of death around the world. High blood pressure is one of the most important risk factors for cardiovascular diseases, leading to organ hypoxia and consequent damage, such as heart failure, stroke, vasculopathy, and nephropathy [4,5,6]. Hypoxic conditions can develop as the result of hypertension-induced alterations in vessel structure and hemodynamics [4,7]. Another cause of tissue hypoxia is abnormal changes in hemoglobin’s (Hb) affinity for oxygen (O_2_), leading to the decreased rate of Hb saturation with O_2_ in the lungs or to the decreased ability of Hb to release O_2_ in peripheral tissues. Thus, the fine tuning of Hb properties ensures the optimal supply of O_2_ to tissue [8,9].

Surface-enhanced Raman spectroscopy (SERS) has been successfully applied to detect different disease biomarkers in blood and other body fluids [3,10,11]. We have proposed an approach for the selective study of the Hb_mb_ in erythrocytes with SERS using a colloidal solution of silver nanoparticles [12] and silver nanostructured surfaces (AgNSS) [13,14]. Plasmonic nanostructures allow us to achieve the enhancement of the Raman scattering from molecules near the nanostructure surface. In the case of erythrocytes, this occurs from membrane-bound Hb (Hb_mb_) bound to AE1-exchanger (also known as band 3 protein). Erythrocyte plasma membrane lipids can be probed simultaneously [15]. Conventional Raman spectroscopy allows one to assess the degree of oxygenation, the conformation, and the affinity of the heme to O_2_ of cytosolic Hb (Hb_c_) in intact erythrocytes. However, it fails to probe Hb_mb_, a minor fraction of Hb, which is more sensitive to oxidative stress, changes in plasma membrane properties, and ion transport through the membrane than Hb_c_ [16,17].

In the present work, we employed SERS with AgNSS to study the effect of spontaneous hypertension on the properties of the plasma membrane and membrane-bound hemoglobin in erythrocytes. The proposed approach is able to detect changes in erythrocyte properties in the early stages of hypertension, such as a decrease in the erythrocyte plasma membrane fluidity and a decrease in the Hb_mb_ heme mobility.

## 2. Materials and Methods

### 2.1. Nanostructure Synthesis

The preparation of AgNSS was performed as described in References [13,18] with modifications. Briefly, 0.3 g of silver nitrate (Sigma, Schnelldorf, Germany, 99.999% purity) was dissolved in 40 mL of MilliQ water under constant stirring, followed by the addition of 30 mL of 20% NaOH solution. The resulting dark precipitate of silver oxide (I) was washed three times with 100 mL of MilliQ water, followed by the addition of 5 mL of 25% aqueous ammonia and 25 mL of MilliQ water, resulting in transparent precursor solution of diamine silver (I) hydroxide. Albedo ultrasonic nebulizer was used to spray the obtained solution of silver complex onto the surface of coverslips placed inside 1000 mL glass on a surface heated up to 340 °C (IKA C-MAG HS 4) for an hour with 5 min breaks every 3–4 min. The coverslips with a silver nanostructured layer were kept at 340 °C for 15 min after the completion of spraying to decompose and desorb possible residual intermediate compounds.

### 2.2. Characterization of Nanostructures

The fabricated AgNSS were visualized by a scanning electron microscope (SEM) NVision 40 (Carl Zeiss). The near-field optical characterization was performed by scattering-type scanning near-field optical microscope s-SNOM “NeaSNOM” (Neaspec, Munich, Germany) based on an atomic force microscope that uses sharp silicon tips covered with a Pt-Ir coating as a near-field probe. The scanning was carried out in a tapping mode with an oscillation frequency of ~250 kHz. The tip–sample interface was illuminated under ∼50° relative to the sample surface by a linearly P-polarized Avesta TiC, Ti:Sapphire continuous tunable laser (700–1000 nm). In this s-SNOM setup, the laser beam is focused on the nanostructures by an upper parabolic mirror. A tip-scattered near-field signal is collected by the same parabolic mirror and goes to a high-sensitive detector, then analyzed after.

### 2.3. Animals

Male 15- or 16-week-old rats from two strains (3 rats in each group) were used in the experiments: spontaneously hypertensive rats (SHR), and Wistar–Kyoto rats (WKY) (BIBCh, Pushchino, Russia). The animal studies were carried out in accordance with the Declaration of Helsinki, EU Directive 2010/63/EU, and the Recommendations of the European Laboratory Animal Science Associations 2014 (FELASA), after permission was granted by the Bioethics Committee of Moscow State University (protocol №82-O, 08.06.2017).

Arterial blood pressure (ABP) was measured noninvasively using the photoelectric plethysmography technique [19] at least 3 times for each rat after adaptation and handling. For WKY rats, the ABP was 139 ± 4 mm Hg; for SHR rats, the ABP was 203 ± 4 mm Hg.

### 2.4. Preparation of Erythrocyte Ghosts

Blood was collected during the decapitation of rats in 15 mL glass tubes containing heparin (10 units/mL). Erythrocyte ghosts (enclosed vesicles consisting of erythrocyte plasma membrane with membrane-bound hemoglobin (Hb_mb_) and submembrane cytoskeleton without cytosolic Hb) were obtained as described in Reference [12]. Briefly, red blood cell mass was hemolyzed in 20 volumes of phosphate buffer (4.7 mM Na_2_ HPO_4_, 0.3 mM NaH_2_ PO_4_ (pH 7.4), 4 °C), followed by three washings and centrifugations (3500 g, 40 min). The erythrocyte ghosts were concentrated by centrifugation (13000 g, 40 min).

### 2.5. SERS Measurements

SERS measurements were performed using an InVia Raman microscope (Renishaw, New Mills, Wotton-under-Edge, Gloucestershire, United Kingdom) equipped with a 20 mW 514 nm argon laser and power neutral density filter (50%). All the spectra were collected using a 20× NA 0.4 objective and 20 s acquisition time. Laser power was 1–5 mM per registration spot. A silicon wafer was used for calibration. The SERS spectra of erythrocytes and erythrocyte ghosts were registered as described previously [12,14]. Before the SERS measurements, suspensions of erythrocyte ghosts were diluted 5 times with the Alen saline (145 mM NaCl, 5 mM KCl, 4 mM Na_2_ HPO_4_, 1 mM NaH_2_ PO_4_, 1 mM MgSO_4_, 1 mM CaCl_2_, pH 7.4). To perform SERS measurements, 300 μL of the suspension of interest were dropped on the AgNSS placed into a glass-bottom Petri dish, then SERS spectra were collected from several places.

### 2.6. SERS Spectra Analysis and Statistics

SERS spectra were processed using open source software Pyraman available at https://github.com/abrazhe/pyraman accessed on 22 November 2021. The baseline was subtracted in each spectrum and ratios of peaks intensities I_1638_/I_1375_, I_1175_/I_1375_ and I_2872_/I_2927_ were calculated after the baseline substraction.

Statistical data were analyzed using GraphPad Prism 8.4 (https://www.graphpad.com accessed on 22 November 2021). The Mann-Whitney U test was used for group comparisons.

## 3. Results

### 3.1. Experimental Design and AgNSS Characterization

Hypertension-related effects were studied in two types of samples obtained from the blood of hypertensive and normotensive rats: on diluted suspensions of erythrocytes and diluted suspensions of erythrocyte ghosts (Figure 1a).

Erythrocyte ghosts serve as a simplified experimental model of the submembrane region of erythrocytes, since they represent enclosed vesicles of the erythrocyte plasma membrane with Hb_mb_, which maintains its interaction with AE1-exchanger [20,21] (Figure 1b). The submembrane region of intact erythrocytes is more complicated and consists of the plasma membrane, Hb_mb_ interacting with the AE1-exchanger, and the cytoskeleton with some amount of Hb_c_ in close vicinity to the inner membrane surface (Figure 1b). To record SERS spectra from molecules in the submembrane region of the studied preparations a small volume of the diluted erythrocyte or erythrocyte ghost suspension was placed on plasmonic nanostructured surfaces (AgNSS in a glass Petri dish). The AgNSS, which were formed by the ultrasonic spraying method [13], have a complex morphology of intersecting circles consisting of silver nanoparticles of different sizes from 5–20 nm up to 100 nm (Figure 2a).

These nanostructures provide a stable and reproducible SERS signal from purified biomolecules, erythrocytes, and mitochondria, as was demonstrated in our previous works [13,18]. Thus, we have already demonstrated the reproducible SERS spectra from Hb_mb_ in erythrocytes and cytochrome *c* in mitochondria recorded from various places on the same AgNSS or from different AgNSS. We also showed the stability of SERS spectra of erythrocytes and mitochondria registered from the same region of AgNSS in time proving the absence of AgNSS degradation in the sample and the absence of the erythrocyte or mitochondria damage by nanostructures [13,18].

Before SERS measurements, the AgNSS were characterized by s-SNOM to visualize the light–sample interaction. The s-SNOM image (Figure 2c) revealed the existence of randomly distributed and strongly localized electromagnetic excitations-hot spots [22,23] that originate mainly due to the gaps between conglomerations of Ag nanoparticles. The comparative analysis of the s-SNOM image (recorded at λ = 720 nm) was performed by directly superimposing one on the corresponding topographical image obtained for a typical conglomeration of Ag nanoparticles (Figure 2b,c). The localization of several hot spots can be clearly seen inside the conglomeration; their appearance should be directly attributed to the gaps between individual Ag nanoparticles. The wavelength used for s-SNOM characterization (λ = 720 nm) is far from the 514 nm argon laser that was used in the SERS experiments. However, this is justified since gap-induced field enhancements are associated with the boundary conditions for the electric field [24,25,26] and therefore weakly depend on the excitation wavelength. Such places on AgNSS with high local electric fields ensure the enhancement of Raman scattering from molecules that are localized not only on the AgNSS, but also on some distance from nanoparticles (e.g., all Hb molecules in the submembrane region of erythrocytes and Hb_mb_ in erythrocyte ghosts).

### 3.2. Spontaneous Hypertension Affects Conformation of Heme in Hb Bound to AE1-Exchanger

The SERS spectra of erythrocytes and erythrocyte ghosts recorded in low-frequency region (600–1800 cm^−1^) originate mainly from the heme molecules of oxyhemoglobin (oHb) (Figure 3a) [12]. The oxygenation of all hemoglobin molecules in studied samples occurred during the high dilution of blood in erythrocyte suspension and during the procedure of ghost preparation. The complete oxygenation of Hb is attributed to SERS peaks that are sensitive to the Hb oxygenation state: the combination of intensive peaks at 1375, 1585, and 1638 cm^−1^ is the signature of oHb, whereas peaks assigned to deoxyhemoglobin (1355, 1555, and 1606 cm^−1^) and methemoglobin (1360–1365, 1565, 1603 cm^−1^) are absent [16,27,28,29]. The peaks at 1638 and 1585 cm^−1^ correspond to vibrations of methine bridges; the peaks at 1375 and 1175 cm^−1^ correspond to symmetric and asymmetric vibrations of pyrrole half rings, respectively; the peak at 1123 cm^−1^ reflects methyl group vibrations. The SERS spectra of erythrocytes and erythrocyte ghosts were stable over time and reproducible for different AgNSSs or different spots on the same region. Appendix A demonstrates SERS spectra of erythrocyte ghosts recorded from three different regions of AgNSS. It can be seen that the spectrum structure and the input of main peaks into the overall spectrum are similar for all the SERS spectra. The same reproducibility was demonstrated with Hb_mb_ in erythrocytes in our previous work [13], showing the absence of the negative effect of the biological sample on AgNSS and vice versa. An important parameter that is usually used to characterize SERS structures is the enhancement factor (EF). EF is calculated from concentrations of the molecule, giving Raman and SERS spectra and the peak intensities of a chosen peak in Raman and SERS spectra. Appendix A demonstrates the Raman and SERS spectra of erythrocytes with a description of the EF calculation for the peak at 1638 cm^−1^ estimated to be ≈ 2.5 × 10^4^.

To investigate the conformational properties of Hb_mb_ heme, we calculated the ratios of the SERS peak intensities: I_1638_/I_1375_ and I_1175_/I_1375_ (Figure 3b,c). The intensity of the peak at 1375 cm^−1^ is used as a normalizing factor, since its intensity does not depend on the heme conformation and can be regarded as a constant value in our experimental conditions. As was demonstrated earlier [30,31,32], the heme *b* in Hb predominantly exists in two conformations: planar and domed [30,31,32]. The first ratio, I_1638_/I_1375_, shows the probability of the planar heme conformation. The higher this ratio is, the higher the probability of a planar conformation of the heme is [33,34]. Moreover, the use of ratios of peak intensities causes the spectral analysis to be independent of the local SERS signal intensity. Our results clearly indicate the absence of the effect of hypertension on the probability of the planar heme conformation (Figure 3b). The ratio of I_1175_/I_1375_ peak intensities represents the probability of the asymmetric pyrrole half-ring vibrations and can be used as a marker of heme mobility: the higher this ratio is, the higher the in-plane mobility of the heme is [33,34]. We found that this ratio is more dispersed in erythrocytes than in erythrocyte ghosts (Figure 3c), which may be explained by the more heterogeneous Hb population in the submembrane region of intact erythrocytes than in erythrocyte ghosts (Figure 1b). We observed no differences in the I_1175_/I_1375_ ratio in the SERS spectra of erythrocytes from normotensive and hypertensive rats (Figure 3c). In contrast, we observed a significant decrease in this ratio for erythrocyte ghosts from SHR rats, indicating the decreased in-plane mobility of the heme in Hb_mb_ (Figure 3c). Moreover, this marker negatively correlates with ABP (Figure 4).

### 3.3. Decreased Fluidity of Plasma Membrane of Erythrocytes under Spontaneous Hypertension

To study the properties of the erythrocyte plasma membrane, we recorded SERS spectra in the high-frequency region (2700–3100 cm^−1^) from erythrocyte ghosts. The SERS spectra of erythrocyte ghosts contain three main peaks at 2872, 2927, and 2962 cm^−1^ (Table 1, Figure 3d), corresponding to =CH_2_ asymmetric, =CH_2_ symmetric, and –CH_3_ group vibrations, respectively [35]. These peaks are known to originate from lipids and proteins. In intact erythrocytes, the submembrane region contains different proteins; therefore, we did not record the SERS spectra of intact erythrocytes. In erythrocyte ghosts, lipids are the main component of the membrane region with the only subpopulation of Hb molecules bound to the AE1 protein. Thus, the SERS peaks at 2872, 2927, and 2962 cm^−1^ are assumed to originate mainly from lipids [15]. The ratio I_2872_/I_2927_ may indicate the ordering of the lipid phase. An increase in this ratio indicates an increase in the number of lipids in trans conformation, which make membrane less fluid [36]. We propose the use of the ratio I_2872_/I_2927_ to evaluate the fluidity of the erythrocyte ghost membrane (Figure 3e). The observed increase in this ratio indicates a decrease in the membrane fluidity. This ratio is statistically significantly higher in SHR rats (Figure 3e), or rats with elevated ABP. This means that spontaneous hypertension leads to the changes in lipid ordering and fluidity in the plasma membrane of erythrocytes.

We also found that the ratio I_2872_/I_2927_ (lipid ordering marker) positively correlates with arterial blood pressure (ABP) in rats (Figure 4a), whereas the ratio I_1175_/I_1375_ (reflecting the in-plane mobility of pyrrole half rings in a Hb_mb_ heme) negatively correlates with ABP (Figure 4b). There is no correlation between the ratio I_1638_/I_1375_ (the probability of the planar conformation of a Hb_mb_ heme) and ABP (Figure 4c). We notice that there is a strong negative correlation between the ratios I_1175_/I_1375_ and I_2872_/I_2927_, showing that the in-plane heme mobility in Hb_mb_ decreases with the decrease in the plasma membrane fluidity.

## 4. Discussion

We applied plasmonic SERS nanostructures to study the properties of the membrane-bound hemoglobin and plasma membrane of erythrocytes and erythrocyte ghosts received from rats with normal and elevated arterial blood pressure. Our results revealed hypertension-induced changes in erythrocyte properties at the molecular level.

It should be noted that high requirements are imposed on SERS nanosensors for biological application–first of all, the reproducibility and stability of the Raman enhancement [37]. The SERS nanosensors used in this study proved themselves to be resistant to the biological preparations and have shown the ability to provide reproducible SERS spectra for a variety of biological objects [13,15,18]. Calculating the ratios of specified SERS peak intensities rather than analyzing absolute peak intensities helps to avoid the influence of variation in signal intensity on the interpretation of the results.

We used the ratio of SERS peak intensities I_2872_/I_2927_ to study the microfluidity of erythrocyte plasma membranes. We revealed decreased membrane fluidity in erythrocyte ghost membranes from hypertensive rats (SHR) compared to healthy normotensive rats (WKY) (Figure 3e), which fits well with clinically relevant data [38].

One of the explanations of this phenomenon is the increased level of cholesterol. An increased level of cholesterol in erythrocyte membranes from people with hypertension and other CVD was shown by Raman spectroscopy and EPR [38,39] and commercial enzymatic assays [40,41]. Increasing cholesterol leads to a decrease in membrane fluidity [42,43] and changes in the erythrocyte function [44]. The SERS ratio I_2872_/I_2927_ was proposed as a marker of cholesterol amount by Faried et al. [45]. Thus, an increase in this ratio, indicating an increase in the membrane stiffness, may be explained by an increased amount of cholesterol in the erythrocyte plasma membrane.

For the first time, we observed a decrease in the in-plane mobility of heme in Hb bound to AE1-exchanger in hypertensive rats (Figure 3c), which is impossible to detect by other methods. Other powerful techniques such as photoacustics and NIRS [46,47] allow determining only Hb_c_ oxygenation, but not other properties. Raman spectroscopy is able to evaluate oxygenation, the affinity of Hb to oxygen, and the conformational properties of Hb_c_, but not Hb_mb_.

We suggest that the detected changes in Hb_mb_ can be explained by the increasing stiffness of the membrane (Figure 4d), which leads to a reduction in the mobility of AE1-exchanger [48], which in turn directly interacts with Hb_mb_ and may affect the conformation and mobility of Hb_mb_ globin. This may reduce the ability of a heme to adapt to changing oxygen concentrations and, thus, to disturb the normal functioning of Hb_mb_ and impair oxygen supply to tissues in hypertensive conditions.

## 5. Conclusions

The use of SERS plasmonic nanosensors allowed us to detect hypertension-induced changes in the plasma membrane and in the conformation of heme in membrane-bound Hb in erythrocytes and erythrocyte ghosts. We revealed a decrease in erythrocyte plasma membrane fluidity and a decrease in the in-plane mobility of heme pyrrole rings in membrane-bound Hb under hypertension, which may affect the affinity of hemoglobin to oxygen. The proposed SERS-based approach may be used to develop novel diagnostic tools to detect early pathologies and to assess treatment outcomes in CVD and beyond.

## Figures and Tables

**Figure 1 biosensors-12-00032-f001:**
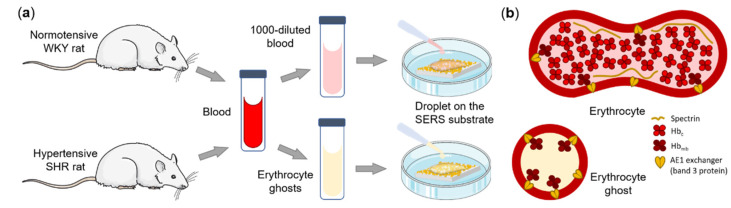
Schematic illustration of (**a**) the experimental design and (**b**) cross-sections of an erythrocyte and an erythrocyte ghost demonstrating the non-homogeneity of hemoglobin molecules in the submembrane region of erythrocytes and the homogeneity of hemoglobin bound to AE1-exchanger in erythrocyte ghosts-enclosed vesicles of erythrocyte plasma membrane.

**Figure 2 biosensors-12-00032-f002:**
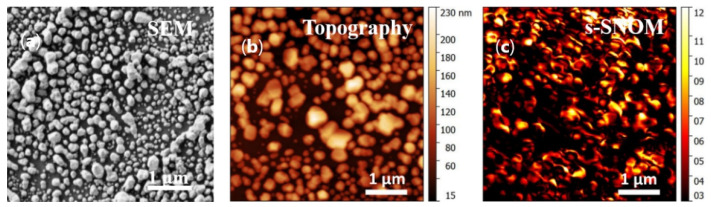
Characterization of plasmonic nanostructured surfaces. (**a**) SEM image of AgNSS; (**b**) topographical (AFM) image of AgNSS aggregates; (**c**) typical pseudo-color s-SNOM image (5 × 5 μm^2^) of Ag nanoparticles obtained at the wavelength λ = 720 nm. The color scale (**c**) shows the optical near-field intensity in arbitrary units.

**Figure 3 biosensors-12-00032-f003:**
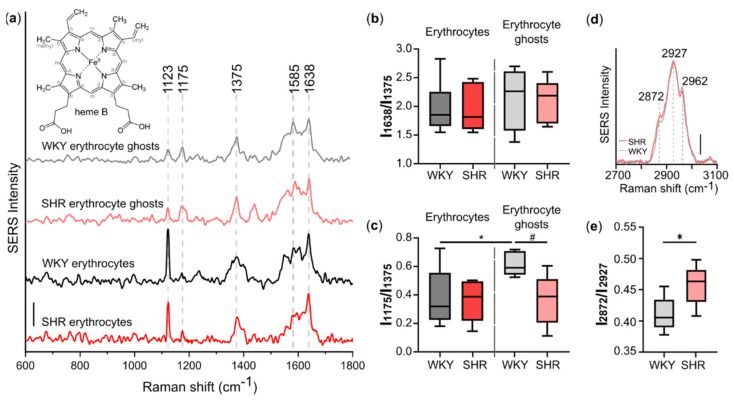
Hypertension-related effects. (**a**) SERS spectra of erythrocyte ghosts and erythrocytes from WKY rats and SHR in the low-frequency region. Spectra are normalized by the intensity of the peak at 1375 cm^−1^. Spectra are shifted vertically for a better presentation. Numbers above peaks correspond to the peak positions (in cm^−1^). Scale 0.5 a. u. (**b**) The ratio I_1638_/I_1375_ (the probability of the planar heme conformation) and (**c**) the ratio I_1175_/I_1375_ (in-plane heme mobility) calculated from the SERS spectra of erythrocytes and erythrocyte ghosts from WKY rats (grey boxes) and SHR (red boxes). * *p* < 0.05; # *p* = 0.0556 (Mann-Whitney test). (**d**) SERS spectra of erythrocyte ghosts from WKY rats (gray) and SHR (red) in the high-frequency region. Spectra are normalized by the intensity of the peak 2927 cm^−1^. Numbers above peaks correspond to the peak positions (in cm^−1^). Scale 0.2 a. u. (**e**) The ratio I_2872_/I_2927_ reflects the ordering of the lipid phase in the membrane of erythrocyte ghosts. * *p* < 0.05 (Mann-Whitney test).

**Figure 4 biosensors-12-00032-f004:**
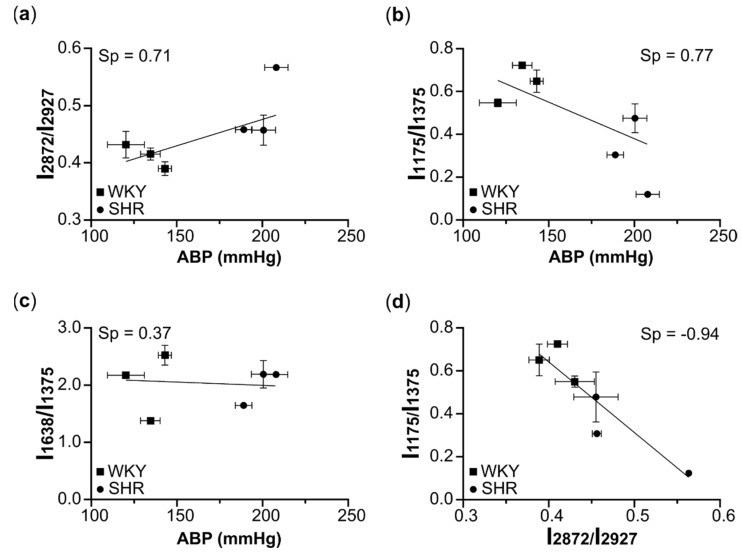
Correlations between different SERS parameters. (**a–c**) Correlations between arterial blood pressure (ABP) (*x*-axis) and the ratios (*y*-axis) of peak intensities in the SERS spectra of erythrocyte ghosts from WKY (squares) and SHR (circles) rats. (**d**) Correlation between the ratios corresponding to in-plane Hb heme mobility and the stiffness of plasma membrane of erythrocyte ghosts. Sp, Spearman correlation coefficient.

**Table 1 biosensors-12-00032-t001:** SERS peak assignment [12,27,28,36].

Peak Position (cm^−1^)	Assignment	Main Contribution from Molecules:
1121	C_b_–CH_3_	HbO_2_
1175	Pyrrole half-ring, asymmetric	HbO_2_
1375	Pyrrole half-ring, symmetric	HbO_2_
1638	C_a_C_m_, C_a_C_m_H, C_a_C_b_	HbO_2_ in planar conformation
2872	ν_as_ ( =CH_2_)	lipids in trans conformation and cholesterol
2927	ν_s_ (–CH_3_)	lipids and proteins
2962	ν_as_ (–CH_3_)	lipids and proteins

## Data Availability

The data presented in this study are available on request from the corresponding author.

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
