# Peer review of "Detection of Hypertension-Induced Changes in Erythrocytes by SERS Nanosensors"

_biosensors, 2022, doi:10.3390/bios12010032_

Round 1

Reviewer 1 Report

The introduction is too much dealing  with Cardiovascular diseases, while the focus of the manuscript is on SERS detection of heme structure and cholesterol content. It appears to me a biochemical/ biophysical study rather than a biosensing issue (I leave to the Editors' policy to judge its appropriateness for publication in Biosensors). I have two main concerns about the results: i) quantitation of SERS signals is a very involved matter when markers from very structurally different systems are followed, since the formation and abundance of hot-spots may differ a lot according to the biological matrice, leading thus to quantitative results scarcely comparable, even if some standardization by using  claimed independent signals is performed (is the hot-spot population quantified?); 2) the cholesterol content, as inferred by the claimed corresponding Raman fingerprint, has to be correlated to well established quantitations by biochemical methods, before  that any statement about its variation inypertension could be made.      

Author Response

Dear Reviewer,

We are very thankful for all the remarks and ideas provided and revised our manuscript significantly according to your comments.

We agree that sensing in cardiovascular diseases is just one of the examples of the potential application of our approach. On the other hand, we think it is important to demonstrate the key points concerning the new area of applications of our SERS sensors including an essay on cardiovascular diseases, since the main focus of modern SERS studies, particularly in terms of their sensor activity, seems to be shifted much to their practical application for complex biological objects [for example, links 3,22-24 in the manuscript]. We revised the Introduction section significantly.

  • The hot spot population is defined mostly by the chemical features of the nanostructured substrate production, while this technique is investigated in detail in many of our previous works listed in the manuscript [links 19-20, 25, 27 in the manuscript]. The key advantage of our nanostructures that it is not a sol of nanoparticles aggregating highly dependent on electrolyte composition and the presence of the biological substance, but rigid nanostructured substrate. Thus, they provide a more stable and reproducible signal from various biological objects, including erythrocytes and their ghosts, which we studied here. It should be noted that biomolecules are not able to change the substrate surface used here and the number and population of hot spots remain the same, which makes it possible to compare our samples in series, as we did in our previous works [19-20, 25, 27 etc.]. Also, a large physical contact area of the cells and cell compartments, compared to the size of hot spot itself, landed onto the substrate would guarantee the same averaged signal from each part of the substrate.
  • We agree that cholesterol content quantification is only one of several possible interpretations of Raman fingerprint – the ratio I2872/I2927 used in the manuscript. More commonly, this SERS ratio is described as a marker of the ordering of the lipid phase [Kutuzov et al. 2011: 10.1088/1612-2011/10/7/075606; Vincent & Levin 1991: 10.1016/S0006-3495(91)82316-5; Gardikis et al., 2006: 10.1016/j.ijpharm.2006.03.023; Gharib et al., 2018 10.1016/j.jddst.2017.12.009], which is broader than just cholesterol content, albeit they are related with each other. This fact does not change the main message of our manuscript, but sounds more correct. We revised significantly this point in the Results and Discussion sections.
  • We have checked and improved the English language and style.

Reviewer 2 Report

The authors show an acceptable level of interesting data. However, the manuscript should address several points as below.

  1. The authors employed the word “plasmic nanosensor” to indicate their technique. It would be helpful if the authors just call it SERS instead of the broad word. 
  2. The authors present a high resolved near-field distribution of nanostructured surface in Figure 2C. However, The wavelength used for s-SNOM characterization is λ = 1600 nm, which is far from the 514 nm as they pointed out. Could authors add the optical characterization results such as extinction spectrum?  It is known semi-random nanostructures can have localized extinction peak and be used for wavelength tunning [DOI:10.1364/OE.20.011466].  Additionally, the localized fields in Figure 2C are found only from relatively bigger particles. 
  3. Could authors provide the information about the number of rats used in experiments?
  4. Authors claim that “spontaneous hypertension in rats leads (i) to an increase in cholesterol content in the erythrocyte plasma membrane and (ii) to a decrease in the mobility of the heme of the membrane-bound hemoglobin.” Could the authors provide more background related to the conventional way of measuring cholesterol and in-plane heme mobility? These can be scientific rationales to use SERS over other ways.

Author Response

Dear Reviewer,

We are very thankful for all the remarks and ideas provided and revised our manuscript significantly according to your comments.

  • The main focus of modern SERS studies seems to be shifted much to their practical application for biosensing [for example, links 3,22-24 in the manuscript], thus we sporadically use the term “plasmonic nanosensor”. We revised the manuscript concerning this issue to clarify, specifying that we used SERS nanosensors.
  • We would like to thank the Reviewer for this comment on s-SNOM characterization. We used s-SNOM mapping to visualize the existence of randomly distributed hot spots, which are strongly localized electromagnetic excitations, in nanoparticles conglomerate. In order to qualitatively demonstrate the presence of hot spots we used a wavelength of 1600 nm, which is the basic laser wavelength in our s-SNOM setup. The near-field signal intensity of hot-spots excitations is mainly determined by the distributions of nanoparticles and enhancement in gaps between nanoparticles slightly depends on the wavelength, but certainly, for shorter wavelengths intensity of the hot-spots should be stronger. But in order to demonstrate it, we have found an opportunity to make measurements at shorter wavelength - 720 nm. Now we have modified Figure 2 in the manuscript by adding the new results obtained by s-SNOM measurements at a shorter wavelength.

(Please, find figs in the attached file)

Figure 2. (a) SEM image of SERS structures (b) Topographical (AFM) image of AgNSS aggregates. (c) Typical pseudo-colour s-SNOM image (4×4 μm2) of Ag nanoparticles obtained at wavelength λ = 720 nm. The color scale shows optical near-field intensity in arbitrary units.

Regarding the linear spectra, it is true, that optical characterization can help wavelength tuning. However, in our case, the reflection spectra recorded by x100 objective from various places of the structure looks are similar to each other and have a wide reflectance dip in the region of 400–850 nm (see Figure). Such light absorption at a wide wavelength range can be explained by a complex hierarchical morphology of our structure. Moreover, the similarity of reflectance spectra recorded from different places is evidence of AgNSS microscopic homogeneity that can be important for serial biomedical studies.

(Please, find figs in the attached file)

Figure. Reflection spectra obtained from fabricated structure.

  • We used 3 rats per group in our experiments, this information is now added to the Methods section.
  • Cholesterol content quantification is just of the possible interpretation of Raman fingerprint I2872/I2927 used in this paper. More commonly this SERS ratio is described as a marker of ordering of the lipid phase [Kutuzov et al. 2011: 10.1088/1612-2011/10/7/075606; Vincent & Levin 1991: 10.1016/S0006-3495(91)82316-5; Gardikis et al., 2006: 10.1016/j.ijpharm.2006.03.023; Gharib et al., 2018 10.1016/j.jddst.2017.12.009], which is confirmed by other methods, first of all EPR studies, allowing the direct measurement of the fluidity of membranes. This explanation is broader than just cholesterol content, albeit they are related to each other. This fact does not change the main message of our manuscript, but sounds more correct. Concerning the heme mobility, Raman and SERS spectroscopy are unique experimental methods to probe in-plane mobility of a heme in liquid phase and especially inside cells. Raman fingerprints for this analysis were evaluated by combining Raman and NMR or X-ray crystallography of different hemes in crystals [Smulevich 2019: 10.1142/S1088424619300088; Sun et al. 2014: 10.1073/pnas.1322274111; Rwere et al. 2007: https://doi.org/10.1002/bip.20887 etc.]. Due to the significant number of previously published papers, we are able to probe heme conformation and mobility of a hemoglobin heme inside living cells by Raman and SERS spectroscopy. The latter provides unique information about membrane-bound hemoglobin in intact erythrocytes, which is impossible to obtain by other techniques. We revised significantly these points in the Results and Discussion sections and added new links.

Round 2

Reviewer 1 Report

The points I have raised have not been substantially addressed.

Author Response

Point 1: The points I have raised have not been substantially addressed.

Response:

With the respect to the Reviewer, all comments are addressed in the present version of the manuscript.

  1. The reproducibility and the stability of used AgNSS that ensure their correct application in the proposed biosensing approach were demonstrated previously by our group on different biological preparations: isolated hemoglobin, erythrocytes, and mitochondria [Semenova et al., J Mat Chem., 2012; Brazhe et al., Sci Rep., 2015]. In the current manuscript, we included SERS spectra of suspension of erythrocyte ghosts recorded from different randomly chosen places of AgNSS (Figure S1, Supplementary Information). The spectra are highly reproducible in their structure and in the relative input of peaks into the overall spectrum demonstrating the stability and the reproducibility of the Raman scattering enhancement of studied biomolecules.

  1. The description of the results about the properties of the plasma membrane of erythrocyte ghosts and the corresponding discussion section was modified. In the revised manuscript we attributed the observed changes in the plasma membrane properties to the decrease in the local membrane fluidity. All corresponding references are included in the manuscript. We did not suggest that our SERS-based approach could be used for the quantitative measurements of the cholesterol content in the erythrocyte plasma membrane. Here we discussed that SERS could "sense" changes in properties of the plasma membrane of erythrocytes indicating several possible reasons. We apologize that after the first revision we did not submit the revised manuscript in the track changes mode that could complicate the finding of the modified and re-wrote manuscript parts.

Reviewer 2 Report

The authors described and revised the most of points. 

Author Response

We are very grateful for the appreciation of our work and all the comments that have improved the manuscript.